

# Radar-derived convective storms climatology for Prut River Basin: 2003–2017

Sorin Burcea[1], Roxana Cică[1], Roxana Bojariu[1]

[1]National Meteorological Administration, Bucharest, 013686, Romania

*Correspondence to*: Roxana Cică (cicadianaroxana@yahoo.com)

**Abstract.** Weather radar measurements are used to study the climatology of convective storms and their characteristics in the transboundary Prut river basin. The Storm Cell Identification and Tracking (SCIT) algorithm was used to processes the volumetric reflectivity measurements, in order to identify, characterize, and track the convective storm cells. The storm attributes table output of the algorithm was used to separate the convective from the stratiform storm cells, by applying a

simple selection criterion based on the average Vertically Integrated Liquid (VIL) values. The radar-derived characteristics of convective storms were used to document the spatial and temporal distributions, and storm properties in terms of duration, traveled distance, movement direction, and intensity. The results show that 94.3% of all convective storm cells were detected during May–August, with the peak in July. The peak time for convective storm cells occurrence was in the afternoon and evening hours between 1000 and 1800 UTC. The average duration of a convective storm was 48.7 min, the average traveled

distance was 28 km, and the average movement speed was 8.5 m s$^{-1}$. The average movement of storms varied with months, but overall most convective storms move from southwest and south-southeast. Also, the analysis shows that the longer-lasting convective storms were the most intense. The spatial distribution of the convective cells reveals yearly variation patterns and hotspots, but also highlights the limitations of radar measurement at longer distances. At the basin scale, the total monthly number of storm cells positively correlates with the precipitation amounts at confidence levels statistically

significant.

## 1 Introduction

Weather events associated to convective storms have large impact on society and natural systems, and can lead to loss of life and property. For instance, large rivers or small catchment flash-floods can cause landslides, while urban areas can be heavily affected if the capacity of the sewer systems is exceeded. During the past decade, heavy rainfalls led to loss of life

and produced significant damages in various areas of Romania. To mitigate the effects of such hazards and to improve the local adaptation to climate change-related natural hazards by increasing the awareness and preparedness of both individuals and stakeholders, knowledge about the spatio-temporal distribution of convective storms of the utmost importance.





A specific challenge is the high variability in time and space of convective weather. Also, regional variations in topography, local and mesoscale flows can also significantly affect the convective storms development. Consequently, high resolution spatio-temporal datasets are needed to perform detailed statistics.

Remote sensing of the atmosphere is nowadays performed on a large scale, instruments like ground-based weather radars
providing high resolution observations of weather systems. Weather radars measurements represent a valuable supplement for thunderstorm and convective events climatology and mapping (Peter et al., 2015; Goudenhoofdt and Delobbe, 2013). Thunderstorm identification and tracking using weather radars are a key tool for severe weather forecasting and warning, and have become, lately, an asset for convective storms climatology as well. Compared to point measurements (i.e., weather stations), the intensity, swath, and spatial extent of the convective events can be derived from radar data.

Lately, based on weather radar data, studies on the statistical characteristics of convective storms around the world have been performed. For instance, thunderstorm climatology in Hungary, using Doppler radar data, has been developed by Seres and Horvath (2015). The study was performed over the period 2004–2012, finding that most storm days occurred in late spring and summer of 2007 and 2010, with the daily frequencies peaking late afternoon. Statistical characteristics of convective storms in Belgium was studied by Goudenhoofdt and Delobbe (2013), using volumetric weather radar observations over a
10-yr period (2002–2011). Their main findings can be summarized to the fact that the probability to observe a high number of storms reaches the maximum in June and in the early afternoon, the motion of convective storms is slower in summer and in the afternoon, while regions with slightly higher convective initiation is related to orography. Over northwestern Italy, a 6-yr warm season (2005–2010, April–September) radar-based analysis of convective storms revealed that storms are more frequent and intense in July and August and during the afternoon hours (Davini et al., 2010). Also, the spatial distribution of
convective initiation highlights that the most suitable regions for the generation of convective cells are located in proximity of hills or mountains, suggesting an active role of the topography in triggering storms.

In Asia, Wang et al. (2014) studied the statistical characteristics of convective initiation in the Beijing-Tianjin region, using radar data over the period 2008–2013. Results showed that there are dense convective initiation activities around the 200-m elevation, meaning that convective storms are more easily triggered over foothills; the highest convective initiation density
was found in the urban areas, while the second highest convective initiation density was found in the forest-type land cover areas. Radar-derived statistics of convective storms and their concomitant changes with thermodynamic variability in Southeast Queensland, Australia, was performed by Peter et al. (2015), concluding that convective storms are found to form and maintain along elevated topography, westerly regime storms occur less frequently and have shorter lifetimes, and that westerly regime storms are primarily driven by large-scale forcing, whereas northerly and trade wind regime storms are more
responsive to surface characteristic. Mohee and Miller (2010) derived a climatology of thunderstorms for North Dakota, using radar and surface thunderstorm data over the 2002–2006 period. It was found that convective storm cells peak in June and July, and in the late afternoon to early morning. Also, their results reveal that the average movement of storm cells varied with months, and the storms moved toward the north, the northeast, and the east.



For Romania, radar-based convective storms identification, observation and their associated damage were also approached, but considering only particular storm events or small periods of time. For example, assessing the severe hailstorm and hail risk of a total of 52 hail events that occurred in May 2013 in southern Romania, using weather radar data, Cică et al. (2015) found that the areas where hail and damage were reported are well captured by the footprints and magnitude of the radar

variables considered in the study. Severe weather situations that occurred in southeastern Romania during 2003–2007 were investigated by Cărbunaru et al. (2013), using weather radar data, their results revealing various Doppler velocity field configurations associated with the intensity of the tropospheric flow, while each configuration of Doppler velocity field was found to be associated to various characteristics of severe convective cells. Lemon et al. (2003) document the strong, long-track, tornado (F3+) that occurred in August 2003, in southeastern Romania, relying on weather radar data for storm

characteristics assessment.

Therefore, even some recent studies on convection and the associated weather events in Romania have been performed, relatively large scale radar-based statistics and mapping of convective storms was not derived. The aim of the research presented in this paper is to derive a climatology of convective storms in the area of Prut River Basin, based on a 15-yr weather radar dataset, to better understand the spatio-temporal characteristics of convective storms that occurred in this

particular area. Also, this study would contribute to the climatology of storms in Europe.

This article is structured as follows. The study area and dataset are described in section 2, and the methodology is presented in section 3. Results are detailed within section 4, followed by the concluding remarks in section 5.

## 2 Study area and data

The area chosen for this study is the Prut River basin (Fig. 1). The Prut River starts from the Forested Carpathian Mountains

in Ukraine, and flows into the Danube River, in Romania. The Prut River is 967 km long, and its catchment area is 27540 km2, with an overall drop of the river of approximately 1500 m. The basin of the Prut River is transboundary, and is located in the territory of three countries: Ukraine, Republic of Moldova, and Romania, with corresponding area percentages of 33%, 28%, and 39% respectively. On the Prut River, the Stânca-Costeşti Dam (located between Stânca (Romania) and Costeşti (Republic of Moldova) was built in the 1970's. The lake serves as reservoir for a hydro power station, but the main

goal of building the dam was to protect villages down the river from floods. In 1970 Romania experienced severe flooding in that area, flooding that repeated in the summer of 2008, when the lake was at 98% of its capacity, and in the summer of 2010.

The Prut River basin is situated in the coverage are of the Weather Surveillance Radar – 98 Doppler (WSR-98D) located at Bârnova, Iași county, Romania (the "+" sign in Fig. 1). The radar is a single-polarization S-band system, operating Nexrad

Volume Coverage Pattern 21 (VCP21), performing a complete volume scan every 6 minutes at 9 different elevation angles. Small parts of the basin, in south and north-west, are covered by another two WSR-98D radars, but the distance from radar location and these areas exceed 200 km. Hence, only data from Bârnova WSR-98D was chosen for this study.



The data used in this study is the output of the Storm Cell Identification and Tracking (SCIT) algorithm (Johnson et al., 1998), covering the period 2003–2017. The SCIT algorithm is a centroid-based storm cell algorithm that processes volumetric reflectivity to identify, characterize, track, and forecast the short-term movement of storm cells identified in the coverage area of the weather radar. The algorithm uses seven reflectivity thresholds (30, 35, 40, 45,50, 55, 60 dBZ) and a

series of adaptable parameters to identify the storm cell centroid. The SCIT is running operationally after each volume scan, resulting tabular, alphanumeric, and graphic products. In this study, the storm attributes table was used, table that contains information like storm ID, azimuth and range, cell movement direction and speed, maximum reflectivity and other.

Although the analysis performed by Johnson et al. (1998) revealed that SCIT detected more than 90% of all storm cells correctly, the algorithm has limitations. Among these, missing the cells close to the radar site because of the cone of silence,

more likely to miss the cells at greater distances because of the poor vertical resolution due to beam spreading. Nevertheless, it is not the scope of this paper to perform an evaluation of the SCIT algorithm.

## 3 Methodology

To reach the goal of the study, first the radar data for the 15-yr period was extracted from the archive, and underwent several steps of processing. The first day with radar data, available in the archive, was 1st of May 2003. For each of the 15years, the

15    days with missing data were identified. The missing days are due to both technical failures of the radar system or to maintenance intervals. One assume that there is no convective activity during the maintenance intervals, as the radar was shut down for maintenance during clear sky days. Also, if during a day more than 10% of data was missing (~ 2 hours), than the day was flagged as missing and excluded from the analysis.

The number of selected days with valid radar data, by year, is illustrated in Fig. 2. The year 2003 has the smallest number of

selected days, as the radar data was available starting with May. The number of selected days per year varies for the interval 2004–2017, being approximately constant for the last 3 years of the period. Thus, from the total number of days excluded from the analysis (13.7% from all calendar days), 3.9% are days with no data, and 9.8% are days with more than 10% data missing. After this step, 4624 days were chosen for the analysis, with approximately 1.1 million scans processed to extract the storm attribute tables.

After extraction of the storm attribute tables, the storm cells selection was performed. The performance of the SCIT algorithm is affected by the distance to the radar, storm located at close ranges are likely to be missed because of the cone of silence, while at greater distances only storms characterized by large vertical development will be identified. Even though according to Johnson et al. (1998), there is some indication that storms located more than 150 km from the radar site are more likely to be missed, and being aware of radar limitations, to have a full coverage of the Prut river basin, storm cells

located within the 20–300 km range interval were kept for further processing.

After the above selection criteria was applied, a total number of just over 2.2 million storm cells resulted, for the whole study period, and for the whole radar coverage area within 20–300 km from the radar. Given the azimuth and range of each of the



storm cell detected, the geographical coordinates (i.e., longitude and latitude) were calculated and included in the storm attributes table to facilitate the mapping process and the extraction of the storm cells within the Prut river basin area. Therefore, the storm cells detected inside the Prut river basin area were extracted from the general data table, resulting in a number of close to 336.000 entries.

In order to keep only the convective storm cells, further selection criteria were applied. When SCIT identifies for the first time a storm cell, a unique name (ID) consisting of letter-number combinations is assigned (e.g., A0), and for the whole time the algorithm detects and tracks that particular ID, the storm attribute table is updated. So, during its lifetime, a unique storm has associated multiple storm cells which are found under the same ID within the table. By looking at the storm attribute tables, we noticed that the algorithm identifies many storms during only one or few consecutive volume scans. According to

American Meteorological Society glossary of meteorology (2018), ordinary convective cells lasts from 20 to 30 min, but often more when forming other types of convective storms (e.g., multicell). To exclude the underdeveloped and fast decaying cells, the condition that a unique storm had a lifetime of at least approximately 20 minutes was applied as further selection criteria. For this, all the unique IDs were identified in the attribute tables, and checked if the storm was identified at least 4 consecutive scans. To ensure that all the storms within the Prut river basin are well captured, the storms that initiated

outside the basin, but during their lifetime were detected at least 4 consecutive scans inside the basin, were retained for analysis.

Based on the cell Vertically Integrated Liquid (VIL) (Greene and Clark, 1972), a supplemental selection was performed to get the final convective storms attribute tables. The VIL (kg m-2) output of SCIT is the radar estimate of the total amount of liquid in the column, for a given storm cell, and this parameter is correlated with updraft strength. Zhang and Qi (2010) used

a single VIL threshold of 6.5 kg m-2, found through subjective analysis of a large number of cases, to separate convective from stratiform rainfall, while Qi et al. (2013) used the same VIL threshold, but within a more complex convective/stratiform segregation algorithm. Within this study, a VIL threshold of 10 kg m-2 was used to separate convective from stratiform storm cells. This value was obtained also through subjective analysis, by arbitrarily checking the storm cell characteristics from different seasons and years. Smaller values of VIL were observed for stratiform cells, shallow

isolated single cells, and for high reflectivity cores (> 45 dBZ) within stratiform precipitation systems, where the radar beam intersects the bright band region, while greater values of VIL (> 10 kg m-2) were observed for convective storm cells. Therefore, if for a unique storm (ID), the average VIL value for its entire lifetime was greater or equal to the threshold value, the storm was retained for analysis, and if the average VIL value was smaller than the threshold value, the storm was discarded. Consequently, a final number of 9507 unique convective storms associated with just over 84000 storm cells were

considered for the analysis over the area of Prut river basin, during 2003–2017.

The mapping of the spatial distribution of the convective storm cells was performed on a 20-km resolution grid, by counting the number of convective storm centroids within each grid cell. The monthly (hourly) convective cells frequency is equal to the average detections in a given month (hour) over the whole 15-yr period. Based on the final convective storm attributes table, 2D histograms comparing radar-derived storm attribute speed, direction of movement, and day of the convective





season were obtained. For each day, and for each direction (i.e., from 1° to 360°) the number of convective storm cells was counted, and plotted on a 2D histogram of 10° x 3-day resolution. A direction of 180° (plotted as south) represents a convective storm cell moving from south to north. The storm speed as function of direction of movement was plotted on a 2D histogram of 5 knots x 10° resolution, illustrating the number of convective storm cell moving with a certain speed on a

given direction.

In addition, the mapping of the spatial distribution of the convective storm cells was also performed on a 10-km resolution grid to match the gridded precipitation data over the Prut basin, for comparison purpose.

Empirical orthogonal function (EOFs) analysis has been applied to the number of convective storm cells spanning the convective seasons (May-August) from 2003 to 2017 to identify spatial and temporal patterns of storm occurrence variability

in the Prut basin.

## 4 Results

### 4.1 Yearly, monthly, and hourly distribution

The annual distribution of the number of convective storm cells detected by the weather radar, within the Prut river basin area, from 2003 to 2017, is depicted in Fig. 3, revealing an inter-annual variation. However, no clear trend is observable, but

a small decrease of the number of convective cells during 2014–2015, followed by an abrupt increase in 2016 when the largest number of convective cells (9766) were detected. The smallest number of cells (2746) corresponds to year 2003, and even if the radar data were available starting with May, we speculate that this number would be representable for this particular year.

The monthly variation of the mean number of the convective storm cells (Fig. 4) detected within the Prut river basin area,

during the 15-yr period, reveals that convection occurs predominantly during the warm season, the frequency being extremely low in the cold season. The convection prevails during May–August, when 94.3% of the total number of convective storms cells were detected, reaching the maximum in July (30.7% of the total number of convective cells).

The mean number of convective storm cells in the Prut river Basin, for the period 2003–2017, was plotted against UTC hours to investigate the diurnal variation of convective cells occurrence (Fig. 5). The majority of convective storm cells

(72.3%) occur in the afternoon and evening hours, being detected between 10 UTC (1300 LT) and 18 UTC (2100 LT), while the rest of the cells were detected with the interval 1900–1000 UTC (2200–1300 LT). Also, during the night hours, convection occurs more frequently than in the morning.

### 4.2 Spatial distribution

Given the results of the monthly distribution of convective storm cells, within the study area and within the 15-yr period

considered for the analysis, the spatial distribution of the convective storm cells was derived only for the warm season months, namely May–August. Also, the mapping of the detected convective cells was performed on the entire coverage area



of the weather radar, to capture the regions more prone to convection, but also to highlight the limitations of the radar measurements.

The spatial distribution of the convective storm cells for each year between 2003 and 2017 (May–August) is illustrated in Fig. 6 and 7, the area of the Prut river basin being also highlighted. Note that even the colour scale is the same, the maximum value is different. Also, for plotting reasons, the distribution was smoothed. First, one observes that the number of convective storms cells detected by the weather radar varies annually. The maps also reveal that, during each of the studied years, a great number of convective events occurred in the mountainous area located west outside the basin, correlating well with the orographic features shown in Fig. 1. Beyond 200 km from radar, the distribution tends to decrease, highlighting the limitation of radar measurements at longer distances.

The annual variability of the number of convective cells detected within the Prut river basin is also revealed in Fig. 6 and 7, the spatial distribution showing different patterns of the areas prone to convection. If, overall, convective cells were detected over the entire area of the basin, there are regions where convection occurred more frequently in some years. Thus, large values of the number of convective cells are observed in the northern and north-eastern sides of the basin in 2003, 2005, and 2007–2008. The spatial distribution is relatively uniform for the years 2006, 2012, and 2017, while in 2004, 2010, and 2009 the convection occurred relatively uniform, but with hotspots over the eastern edge and southern and northern parts, respectively. Convection occurred more frequently in southeast and northeast during 2011, while in 2015–2016 isolated convection hotspots are observed in southern part of the basin. In 2013–2014, large number of convective storms were detected over the southern, central, and northeastern areas of the basin. The smallest value of the number of convective storms is depicted over the northwestern part of the basin, for the entire 15-yr period, and this pattern could be explained by the limitation of detection at longer distances from the radar location. At this distances, only very intense and strong vertical developed storms are detected.

### 4.3 Storm properties

Various storm characteristics have been deducted from the analysis of the convective storms detected within the Prut river basin between May and August, 2003–2017. So, considering the 94.4% of all the 9507 convective storms detected by the weather radar, storm average duration, average distance traveled, average movement speed, average maximum reflectivity, and average VIL were calculated. The yearly average values of these parameters are shown in Table 1. One observe that the average duration of storms is larger in the first four years of the studied period (i.e., 2003–2006), with a maximum of 52.1 min in 2006, followed by a decrease as the period progresses, reaching a minimum of 45.7 min in 2017. The average traveled distance and movement speed of a convective storm peaks in 2008 (34.7 km and 11.4 m s-1), the minimum of 24.9 km and 7.4 m s-1, respectively, being characteristic of the year 2015. The average storm intensity in maximum reflectivity and VIL also presents yearly variations, with a maximum in 2016 (55 dBZ and 23 kg m-2) and a minimum in 2003 (51 dBZ and 16 kg m-2).



For all convective storms, over the 15-yr period, the average duration was found to be 48.7 min. For all storms with a duration of more than 30 min, the average was found to be 61.9 min, while for all storms with a duration of more than 1 h, the average was discovered to be 88.1 min. Although the average duration of storms was less than 60 min, the longest-lasting convective storm in 2003–2017, within the basin, had a duration of 368 min (6 h 8 min) and was recorded in August 2016.

The duration of the longest-lasting convective storm in a particular month between 2003 and 2017 was found to be 242, 215, 235, and 368 min in May, June, July, and August, respectively. It was also found that convective storms starting to develop between 0600 and 1000 UTC (0900 and 1300 LT) had, in average, longer durations (52.7 min) than the storms originating at the remaining hours (48.4 min). Of all the convective storms, it was found that 65.8%, 26.9%, 3.2%, and 0.4% had duration of more than 30 min, 60 min, 120 min, and 180 min, respectively.

For all storms, the average traveled distance was found to be 28 km, while the longest track was 224 km. Of all the storms, only 10.6% were both more than 60 min in duration and more than 50 km in traveled distance. The average maximum reflectivity and VIL values of these convective storms were 54.4 dBZ and 22.0 kg m-2, respectively. They were not much larger than the overall average of 53.4 dBZ and 20.2 kg m-2, respectively. These storms have an average movement speed of 10.8 m s-1, which was higher than the overall average of 8.5 m s-1. For storms that were more than 60 min and more than

100 km long, the maximum reflectivity and VIL values were 55.0 dBZ and 23.8 kg m-2, respectively, which were little larger than those described earlier. The average movement speed increases to 13.4 m s-1, higher than the overall average above. From this analysis, also resulted that the convective storms with average reflectivity < 50 dBZ had a duration of 41.3 min and 13.1 kg m-2, while those with average reflectivity > 55 dBZ had a duration of 53.2 min and 29.5 kg m-2. Consequently, it can be concluded that a relation between the storm duration and the storm intensity exists, the longer-lasting

storms being more intense.

The frequency of occurrence is related to the intensity of severe weather events, approximately following a log-linear decrease with increasing intensity (Brooks and Doswell, 2001; Brooks and Stensrud, 2000). Based on this approach, the average storm reflectivity, average VIL, average duration, average movement speed, and average traveled distance values were divided into groups (Table 2) and plotted against their percentages from the total number of convective storms detected

between May and August (Fig. 8). For example, the graph shows a decrease of the number of convective storms with increasing reflectivity. Of the total number of convective storms detected between May and August 2003–2017 (8984), 69.95% (6284 storms) are associated with an average maximum reflectivity between 40 and 55 dBZ, 25.24% (2268 storms) are associated with an average maximum reflectivity between 55 and 60 dBZ, while 4.81% (432 storms) of all convective storms had the average maximum reflectivity equal to or greater than 60 dBZ. This behaviour is characteristic to all the

investigated parameters characteristic to convective storms, their distribution revealing that the frequency of occurrence of intense convective storms tends to follow an approximately log-linear decrease with increasing intensity, implying that weaker convective storms are more often occurring than stronger convective storms. This finding is consistent with the results of other studies (e.g., Goudenhoofdt and Delobbe, 2013; May and Ballinger, 2007).



The 2D histogram comparing radar-derived direction of movement of storm cells and day of the convective season (between May and August 2003–2017), for the Prut river basin (Fig. 9) shows that the direction of movement varies each month. Moreover, the histogram reveals that there are daily intervals within a month when a great number of storm cells travel on preferential directions. In May, the greatest number of convective storms cells occurred during the second half of the month, the largest number of storm cells moving from south-southeast. During June, convective storms occurred, overall, all days of the month, with the maximum number of storm cells moving from southwest. July is characterized by the greatest number of convective storm cells, as depicted by the monthly distribution as well (Fig. 4), the largest number occurring during the last third of the month. In this case, the preferred direction of movement was from southeast, east-southeast, and southwest. In August, the majority of convective storms cells occurred during the first days and at the beginning of the second half of the month. For the former period, the movement of most of the cells was from southwest and east-southeast, while for the latter period the movement was preferentially from southeast.

Overall, during May–August, from the total number of unique convective storms (8984), 12.2% (1099 storms) had an average movement direction from southwest (between 210° and 240°), and 10.8% (967 storms) had an average movement direction from south-southeast (between 140° and 170°). The maximum of 4.72% (424 storms) was found on the direction between 230° and 240°, while the minimum of 0.36% (33 storms) was found on the direction between 0° and 10°.

The 2D histogram comparing radar-derived direction of travel of storm cells and average storm cells speed (Fig. 10) is consistent with the histogram in Fig. 9 in terms of storm cells movement direction, revealing that the lowest number of storm cells had a movement direction generally from northwest. From all the convective storm cells detected within the Prut river basin, between May and August (2003–2017), 80.4% had an average movement speed less than or equal to 12 m s-1. Most of the convective storm cells moving on a general direction from northeast and southwest, had average speeds between 3 m s-1 and 9 m s-1, while those moving on a general direction from southeast had average speeds between 6 m s-1 and 12 m s-1. The largest number of convective storm cells with average moving speed between 12 m s-1 and 18 m s-1 is observed on the general directions between southeast and southwest, respectively.

## 4.4 Relation between basin-averaged number of storms and observed precipitation

The relation between the basin-averaged number of storm cells and precipitation amount for the months in the active season (May to August) has been investigated using gridded precipitation at different spatial resolutions. The CRU precipitation dataset (Harris et al., 2014) has a spatial resolution of 50 km in latitude and longitude. Two other precipitation datasets (with 10- km resolution and 1-km resolution in latitude and longitude) have been also used. They were developed for the Prut basin in the IMDROFLOOD project (http://imdroflood.meteoromania.ro/geoportal/). The correlation coefficients between basin-averaged number of convection cells and precipitation amounts are positive and statistically significant for all gridded precipitation datasets, but the highest magnitudes are for the datasets with 10 km-resolution and 1-km resolution (Table 3 and Fig. 11).





Even though the time interval is limited to 60 months (15 seasons) it is interesting to note the downward trend in the basin-averaged ratio of precipitation amounts and number of storm cells (Fig 12). This signal is related to the upward trend in the number of convective cells associated with a downward trend in precipitation amounts. Several physical mechanisms shape the extreme precipitation response to thermodynamic and dynamic factors. The thermodynamic contribution is relatively

well understood, but theoretical understanding of the microphysical and convective-related dynamical contributions is still under development (O'Gorman, 2015). It is not the purpose of our study to analyze in depth such issues. However, we want to highlight here the importance of integrating radar measurements with other types of meteorological observations to better document the risks related to extreme precipitation in the context of climate variability and change.

## 4.5 Spatial and temporal patterns of variability in number of storms

The 1st EOF of anomalies in the seasonal sum (May-August) of storm cells number shows an overall increase in convective cells in the period 2003-2017 over the Prut basin (Fig. 13). The largest magnitude of the increase is in the central and lower part of the river basin. This pattern is consistent with the upward trend detected in the basin-averaged number of storm cells for the months of the active season (May to August) (Fig 11).

       The 2nd EOF shows a more local feature: the opposite behaviour of seasonal storm cell number over the north/northeastern

(the red-shaded area in Fig. 14) and central/southern part of the basin (the blue-shaded area in Fig. 14). The north/northwestern area of the basin shows a weak signal in both EOF 1 and EOF 2 spatial configurations suggesting that in that region there are limitations of radar measurement at longer distances.

## 5 Conclusions

       The characteristics of the convective storms within the transboundary Prut river basin have been analyzed over a 15-yr

period (2003–2017). The analysis is based on radar data, and the SCIT algorithm has been used to detect and track convective storms on successive 6-min volumetric scans. More than 2.2 million storm cells were detected in the coverage area limited to between 20 and 300 km from the radar, for the whole study period. A simple selection criterion, based on the average VIL values, has been used to separate convective from stratiform storm cells. A number of 9507 convective storms, associated with just over 84000 storm cells and a duration of at least approximately 20 minutes, have been identified in a

study area.

       From the analysis of the results in this study, a few matters of the climatology of convective storms in the Prut river basin became clear. A yearly variation of the convective activity was observed, the largest number of convective cells being detected in 2016. The monthly variation revealed that the convection prevails during May–August, when more than 90% of the convective cells were identified. According to the same analysis, the peak was registered in the month of July, when

close to one third of the total number of convective cells were detected. The majority of convective storm cells occurred in



the afternoon and evening hours (between 1000 UTC and 1800 UTC), while during the night, convection occurred more frequently than in the morning.

Although the spatial distribution of the convective cells shows that convection developed throughout the basin area, yearly patterns are observed. The analysis revealed convection hotspots over the northeast, southeast, or the eastern edge of the basin. The northwestern part of the basin has been identified as the region were the smallest number of convective cells were detected, but this could be explained by the limitations of radar measurements at long distances. Nevertheless, despite the inherent limitations of radar measurements, this type of analysis offers one of the most complete information on the convective storm's characteristics in the coverage area.

Considering the convective storms that occurred between May and August (2003–2017), their radar-derived characteristics have been analyzed. The average duration of a convective storm was found to be 48.7 min, while the average traveled distance was found to be 28 km. The average movement speed of a storm was 8.5 m s-1, increasing to 13.4 m s-1 for the convective storms that were more than 60 min and more than 100 km long. It was also found that the longer-lasting convective storms were more intense, and their frequency of occurrence approximately follows a log-linear decrease with increasing intensity.

The convective storms movement direction and speed analysis highlighted the fact that there are intervals within a given month when a great number of storm cells travel on preferential directions. Overall, during May–August (2003–2017), from the total number of convective storms, 12.2% had an average movement direction from southwest, followed by 10.8% from south-southeast. Also, 80.4% had an average movement speed less than or equal to 12 m s-1, while the largest number of convective storms moving with speeds between 12 m s-1 and 18 m s-1 is observed on the general directions between southeast and southwest, respectively.

This study, which is based mainly on radar data, offers perspective for better understanding of convective storm activity, which is of paramount importance in assessing the risks associated with extreme precipitation and hail episodes in the context of climate variability and change. First, given a study area and the density of radar sites, additional observations from neighbouring radars should be included to improve the quality of the observations in the overlapping areas. Supplemental storm properties could be analyzed by including other datasets like lightning, satellite and ground observations. The latter could be also used to assess the performance of the convective storms' identification and tracking algorithm, and, by this, to derive comprehensive and high-resolution climatology of weather events such as extreme precipitation and hailfall. Also, our study aims at drawing the attention to the importance of integrating radar measurements with other types of meteorological observations to better document the risks related to extreme meteorological events in the context of climate variability and change.

**Author contribution**

All the authors contributed equally to the manuscript.



**Competing interests**

The authors declare that they have no conflict of interest.

**Acknowledgements**

The authors would like to thank the European Union and Romanian National Authority for Scientific Research and
Innovation (UEFISCDI) for funding, in the frame of the collaborative international consortium IMDROFLOOD financed
under the ERA-NET Cofund WaterWorks2014 Call. This ERA-NET is an integral part of the 2015 Joint Activities
developed by the Water Challenges for a Changing World Joint Programme Initiative (Water JPI). This work was supported
by the grant of the UEFISCDI, project number 81/2016, within PNCDI III. Also, we thank our colleagues Alexandru
Dumitrescu and Sorin Dascălu, from National Meteorological Administration of Romania, for their help on the precipitation
and model datasets.

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



**Table 1 Characteristics of convective storms in Prut river basin, for different years of the 2003–2017 period.**

| Year | Average duration (min) | Average traveled distance (km) | Average movement speed (m s⁻¹) | Average maximum reflectivity (dBZ) | Average VIL (kg m⁻²) |
|---|---|---|---|---|---|
| 2003 | 51.3 | 27.3 | 7.6 | 51 | 16 |
| 2004 | 49.4 | 30.1 | 8.9 | 53 | 20 |
| 2005 | 51.2 | 26.6 | 7.6 | 53 | 20 |
| 2006 | 52.1 | 29.3 | 8.3 | 53 | 20 |
| 2007 | 48.7 | 29.9 | 9.3 | 53 | 21 |
| 2008 | 48.1 | 34.7 | 11.4 | 54 | 22 |
| 2009 | 49.1 | 29.1 | 8.7 | 53 | 20 |
| 2010 | 46.8 | 26.7 | 8.4 | 52 | 19 |
| 2011 | 48.8 | 25.5 | 7.4 | 52 | 19 |
| 2012 | 46.9 | 28.2 | 8.9 | 54 | 19 |
| 2013 | 48.4 | 26.8 | 7.8 | 55 | 20 |
| 2014 | 49 | 27.5 | 8.1 | 53 | 16 |
| 2015 | 47.6 | 24.9 | 7.4 | 53 | 18 |
| 2016 | 49.2 | 26.5 | 7.8 | 55 | 23 |
| 2017 | 45.7 | 26.9 | 8.7 | 54 | 23 |



**Table 2 Radar-derived parameters of convective storms, range of values and their associated percentages from the total number of convective storms detected within Prut river basin.**

| | Group no. | Group 1 | Group 2 | Group 3 |
|---|---|---|---|---|
| Reflectivity (dBZ) | Values range | $40 \leq dBZ < 55$ | $55 \leq dBZ < 60$ | $60 \leq dBZ$ |
| | Percent | 69.95% | 25.24% | 4.81% |
| VIL (kg m$^{-2}$) | Values range | $10 \leq kg\ m^{-2} < 20$ | $20 \leq kg\ m^{-2} < 40$ | $40 \leq kg\ m^{-2}$ |
| | Percent | 60.47% | 34.74% | 4.79% |
| Duration (min) | Values range | $20 \leq min < 60$ | $60 \leq min < 120$ | $120 \leq min$ |
| | Percent | 71.05% | 25.63% | 3.32% |
| Speed (m s$^{-1}$) | Values range | $0 \leq m\ s^{-1} < 10$ | $10 \leq m\ s^{-1} < 20$ | $20 \leq m\ s^{-1}$ |
| | Percent | 68.10% | 29.63% | 2.27% |
| Distance (km) | Values range | $0 \leq km < 30$ | $6 \leq km < 60$ | $60 \leq km$ |
| | Percent | 65.44% | 26.99% | 7.57% |





**Table 3 Correlation coefficients between basin-averaged number of storms and precipitation amounts for the months in the active season (May to August) of the time interval 2003-2017.**

|  | CRU Prut precipitation (50 km -resolution) | Prut precipitation (10 km -resolution) | Prut precipitation (1 km -resolution) |
|---|---|---|---|
| Correlation Coefficient | 0.57 | 0.60 | 0.60 |



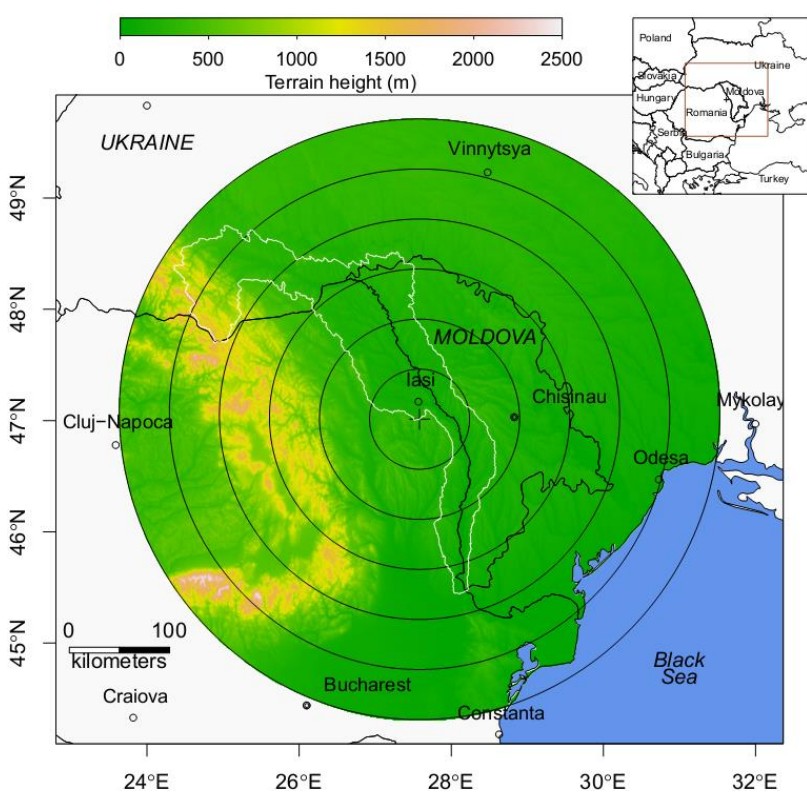

**Figure 1: Map of the study area, showing the Prut River basin (white polygon), Bârnova WSR-98D radar location ("+" sign), orography and borders. The range rings around the radar are 50 km apart, extending to 300 km.**



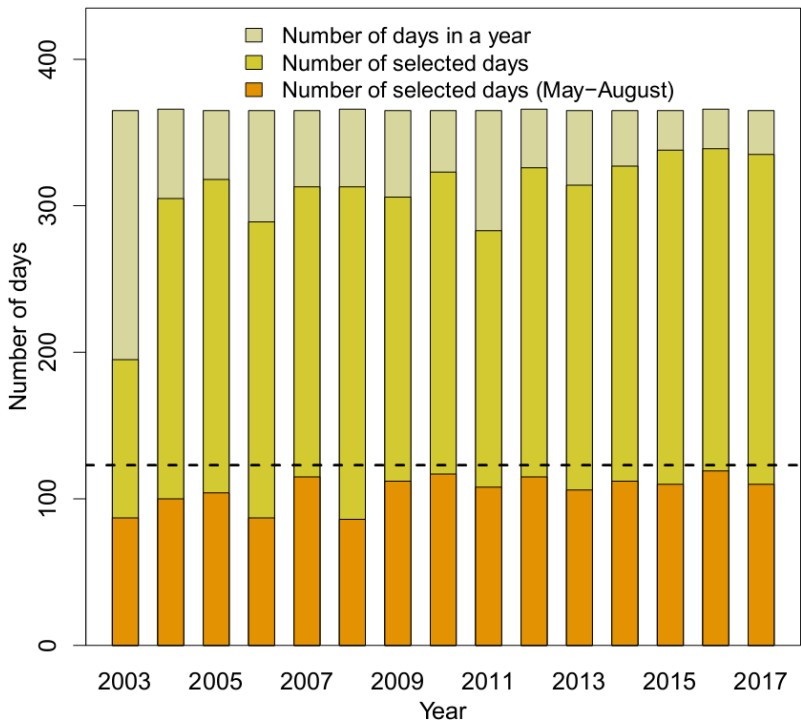

**Figure 2: Number of days per year, within the period 2003–2017, with valid data selected for the analysis. The number of selected days between May and August is also depicted, the dotted line representing the number of days (i.e., 123 days) from 1st of May to 31st of August.**





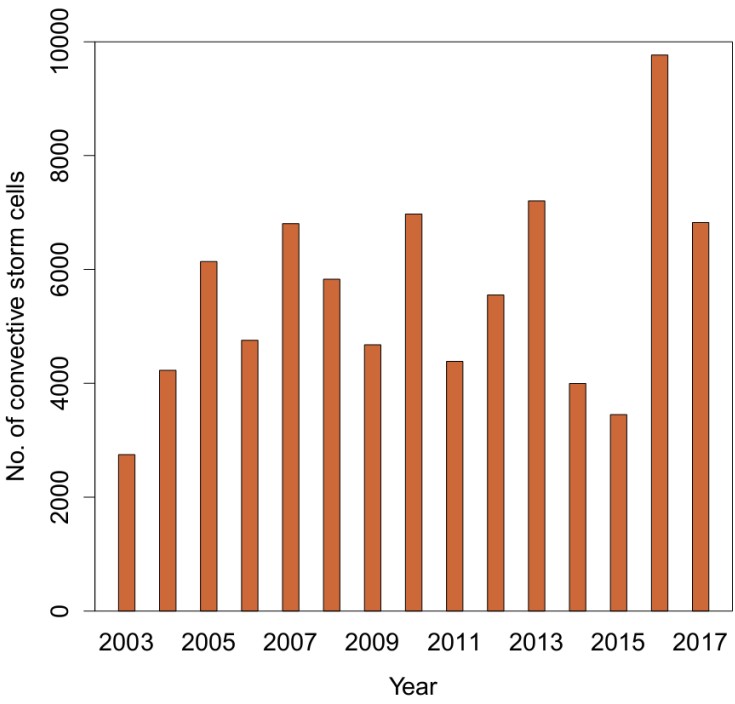

Figure 3: Annual distribution of the number of convective storm cells in the Prut river Basin, for the period 2003–2017.

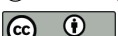



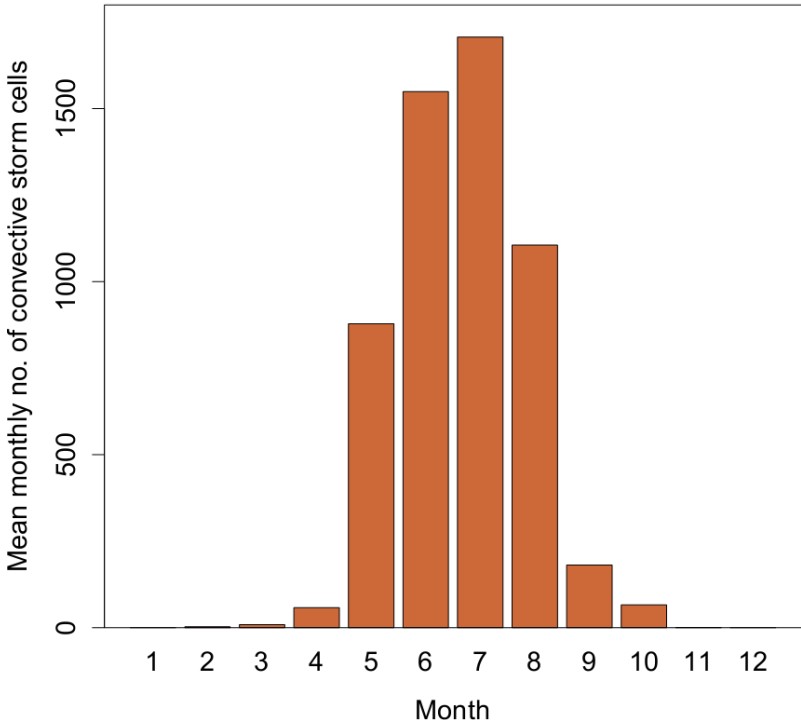

**Figure 4: Monthly distribution of the mean number of convective storm cells in the Prut river Basin, for the period 2003–2017.**





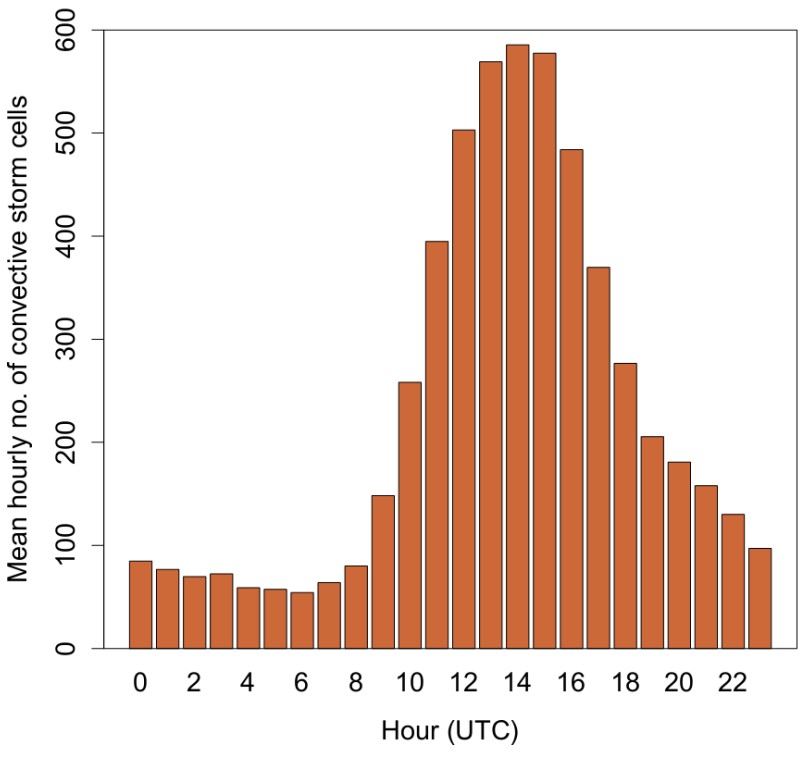

**Figure 5: Hourly distribution of the mean number of convective storm cells in the Prut river Basin, for the period 2003–2017.**

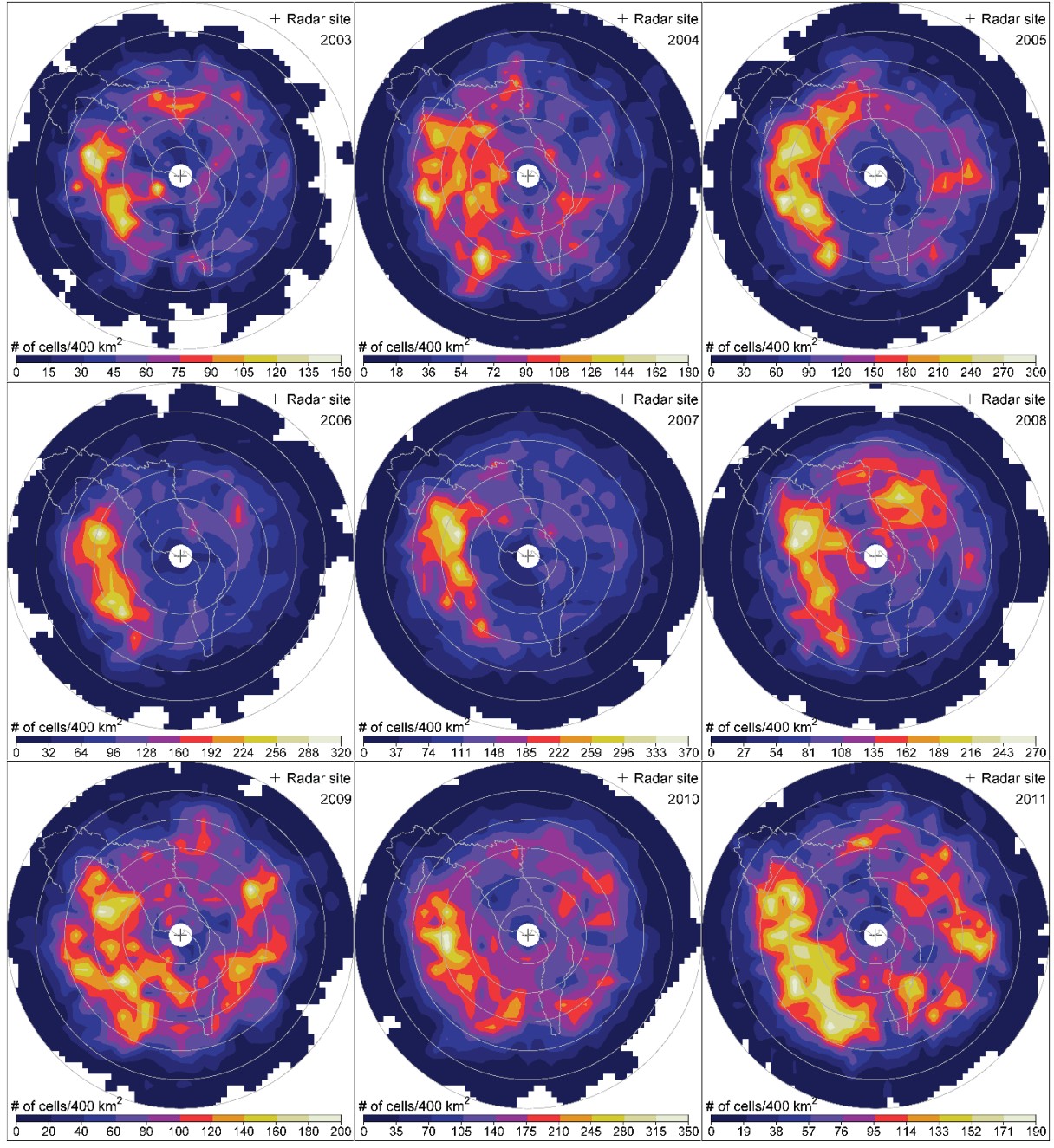

**Figure 6: Spatial distribution of the convective storm cells in the study area, for the period May–August, for each year between 2003 and 2011. The year is denoted in the upper right corner of each panel, starting from top to bottom and form left to right.**





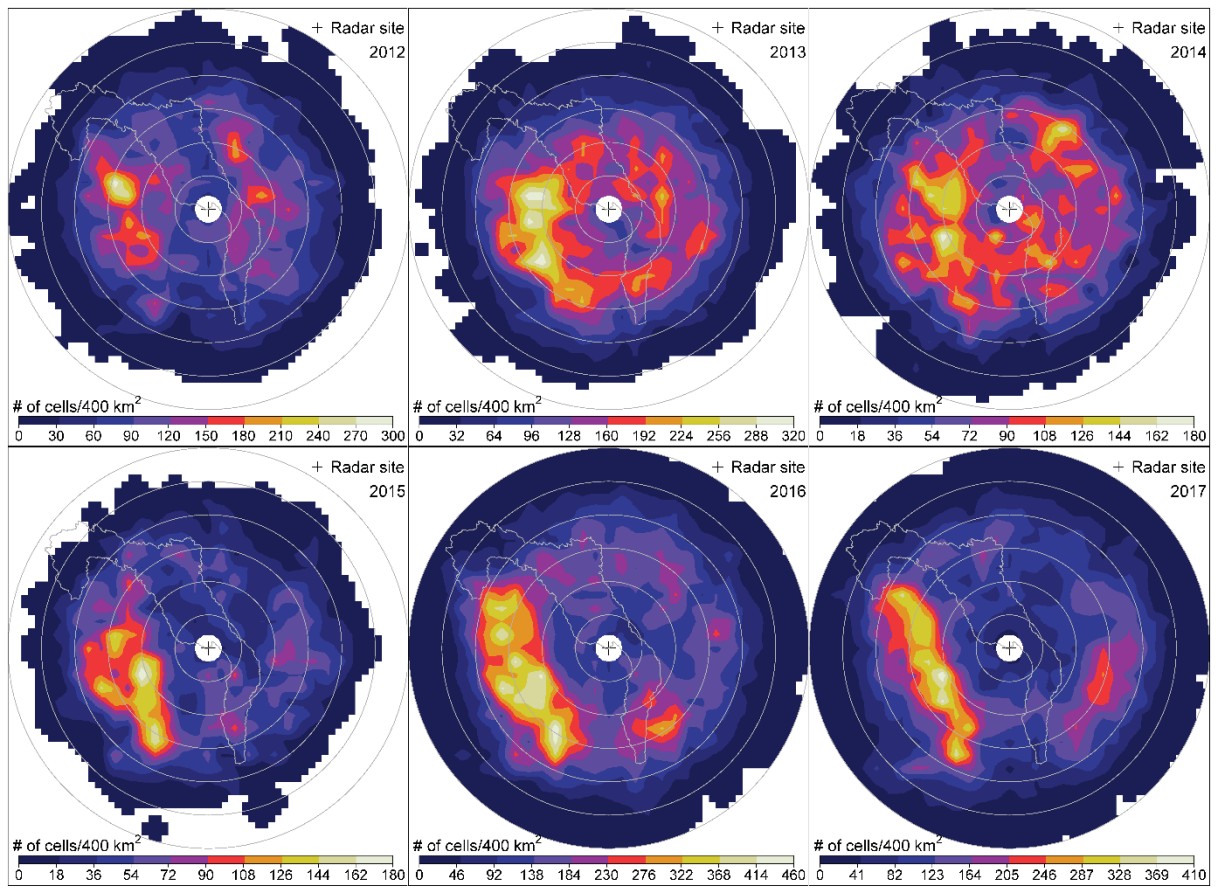

**Figure 7: Same as Fig. 6, but for the period 2012–2017.**





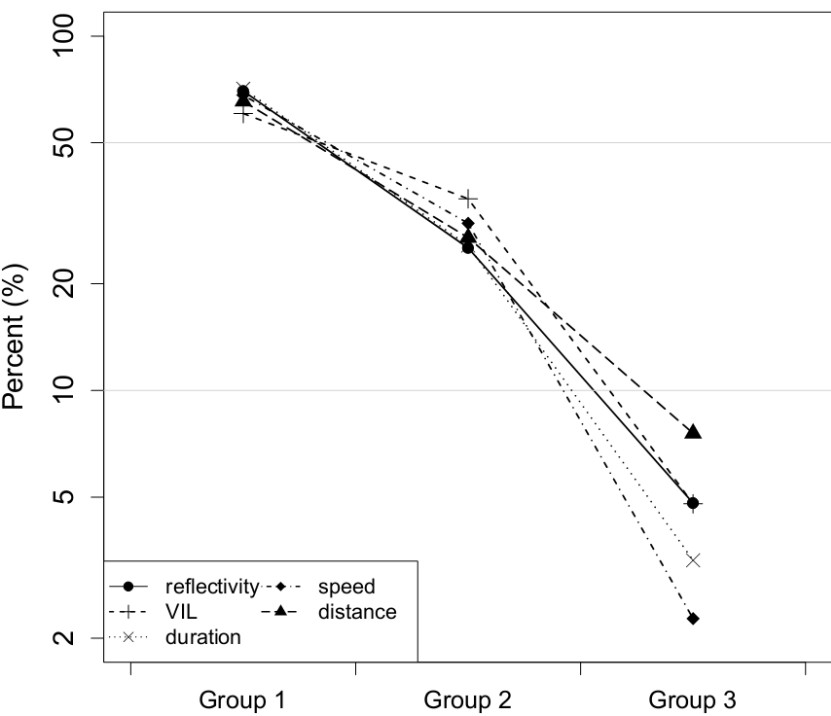

**Figure 8: Distribution of parameters of convective storms as function of their percentages.**





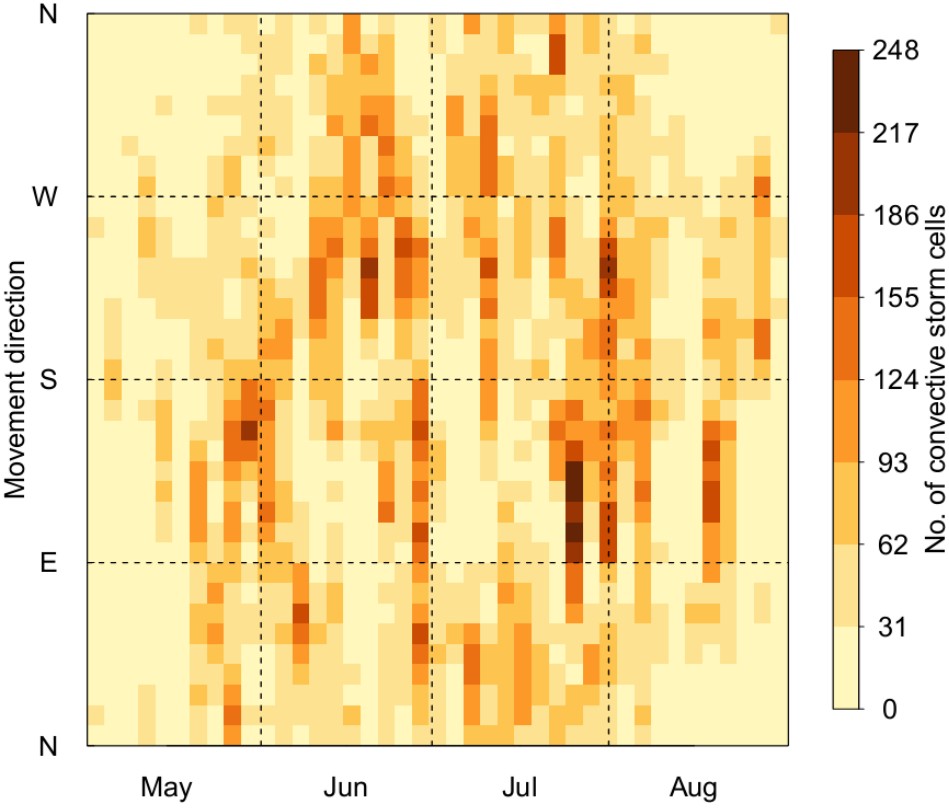

**Figure 9: 2D histogram of the number of convective storm cells as function of month and movement direction, for the Prut river basin, between 2003 and 2017 (May–August).**





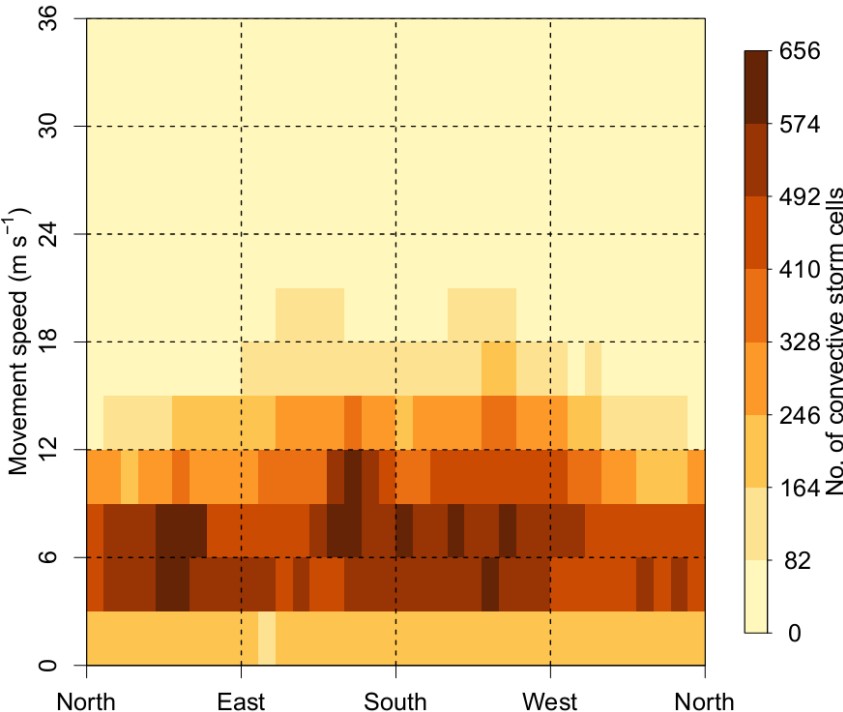

**Figure 10: 2D histogram of the number of convective storm cells as function of movement direction and storm speed, for the Prut river basin, between 2003 and 2017 (May–August).**





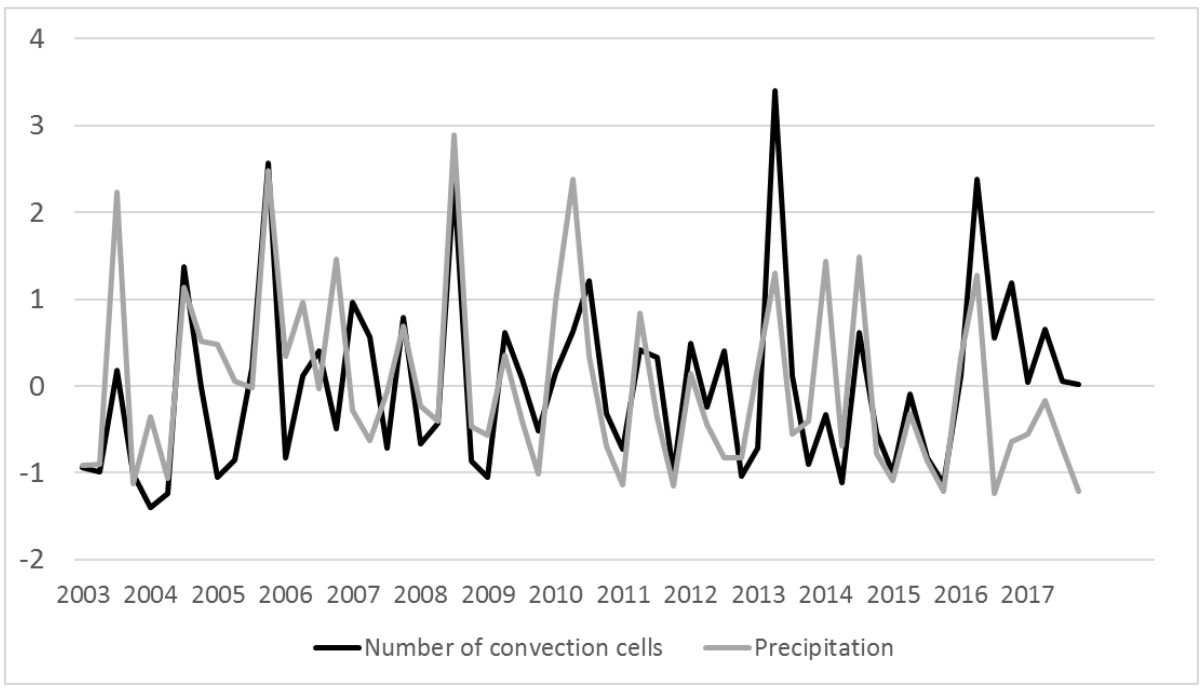

**Figure 11: Basin-averaged standardized anomalies of number of storm cells and precipitation amounts for the months of the active season (May to August) in the period 2003-2017. The precipitation amounts averaged over the Prut basin are computed from the gridded dataset with 10-km resolution in latitude and longitude. The correlation coefficient is 0.60.**





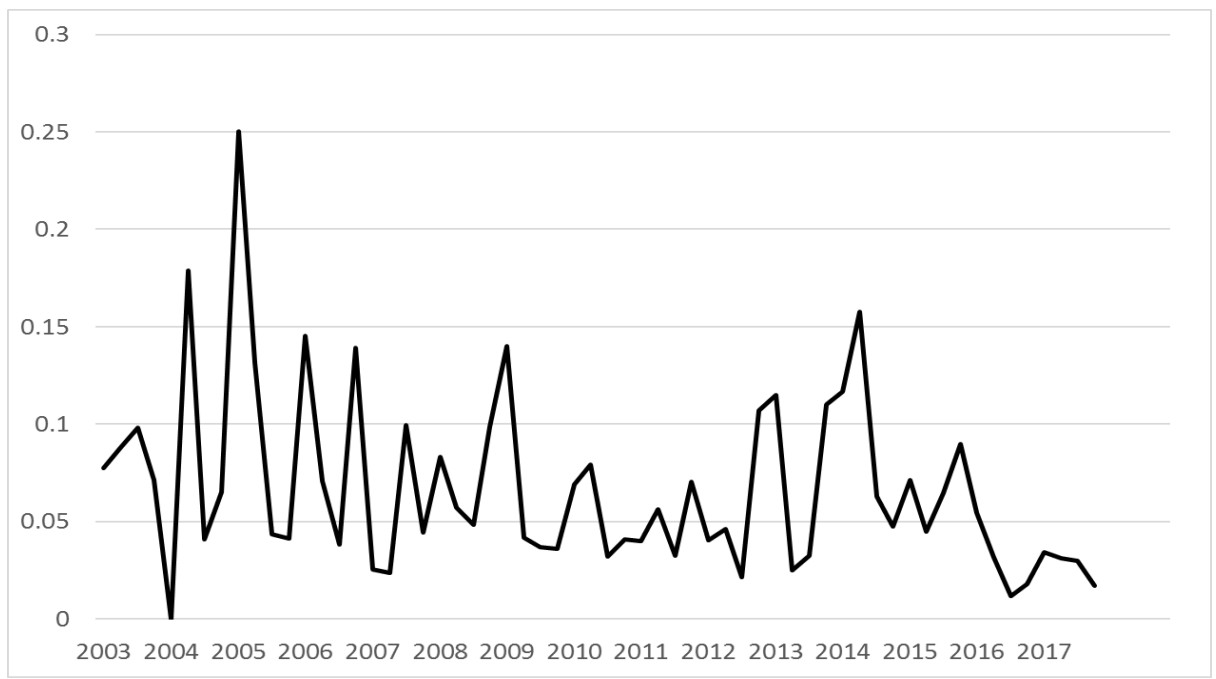

**Figure 12: Monthly ratio of basin-averaged precipitation (in mm) and number of storm cells for the months of the active season (May to August) in the period 2003-2017.**





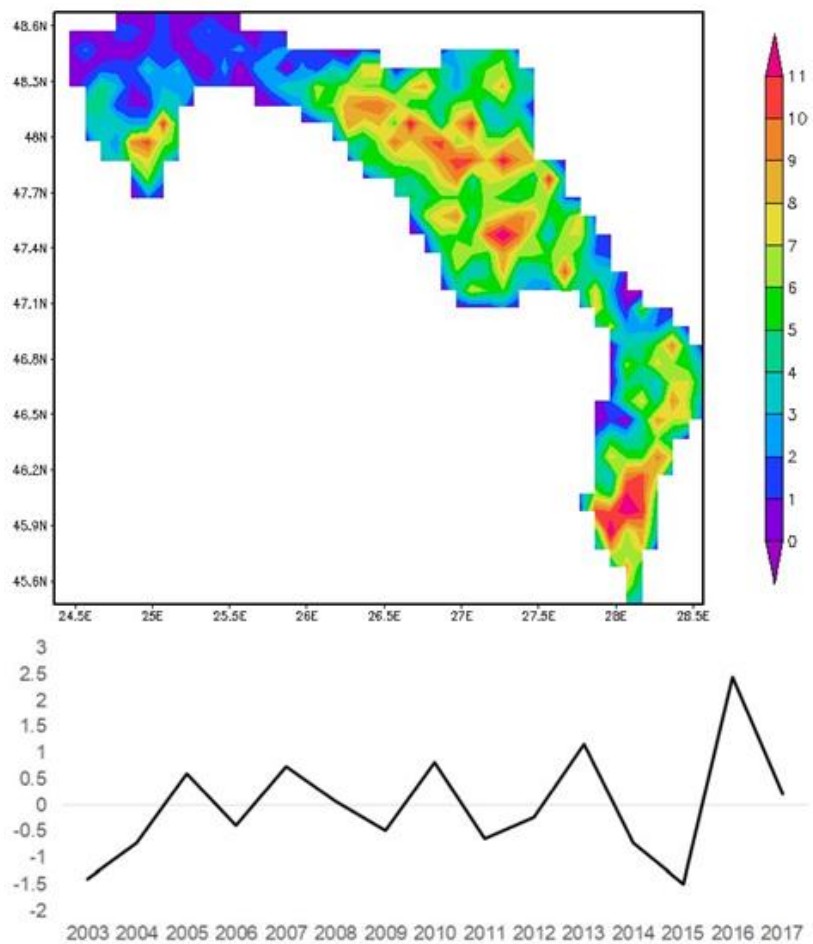

**Figure 13: Spatial configuration (upper panel) and the associated principal component (bottom panel) of the 1st EOF mode for the seasonal sum (May-August) of storm cells number in the period 2003-2017 over the Prut basin.**

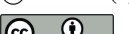


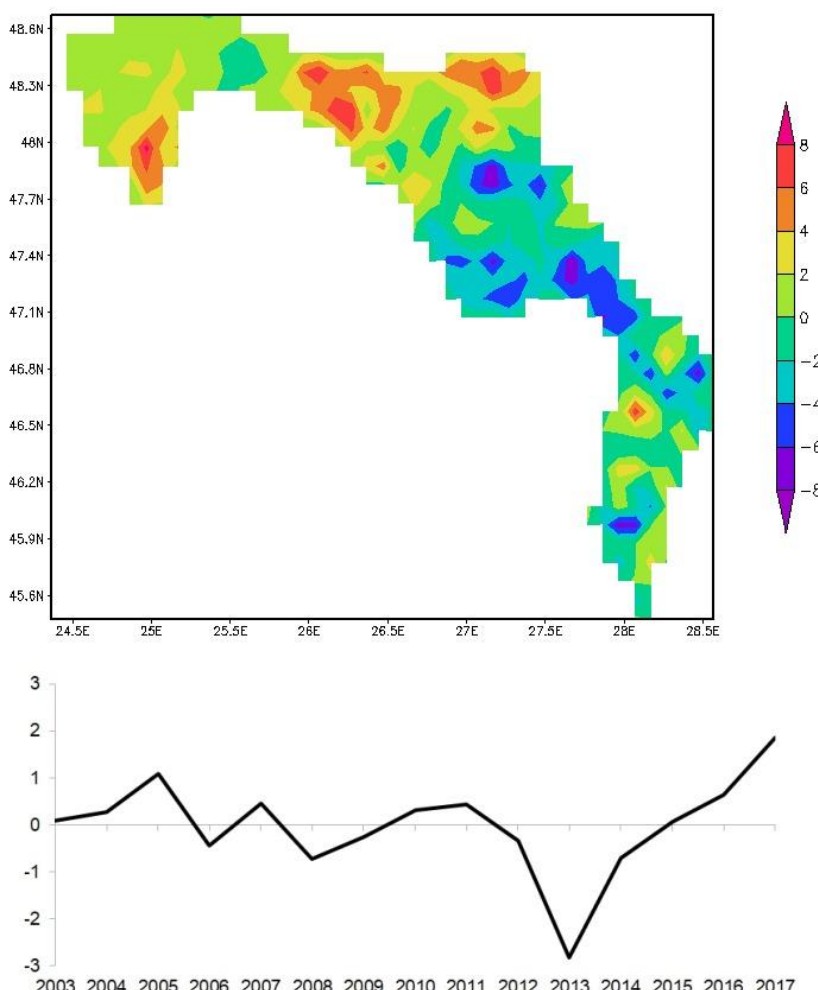

**Figure 14: Spatial configuration (upper panel) and the associated principal component (bottom panel) of the 2nd EOF mode for the seasonal sum (May-August) of storm cells number in the period 2003-2017 over the Prut basin.**

