# Peer review of "Radar-derived convective storms climatology for Prut River Basin: 2003–2017"

_Natural Hazards and Earth System Sciences, 2018_

## Referee Comment (RC1) · Anonymous Referee #1 · 13 Jan 2019

The manuscript investigates convective activity 15-years climatology in Prut basin, Romania analysing a dataset derived from Storm Cell Identification and Tracking (Johnson et al., 1998) applied to the WSR-98D weather radar located in Barnova, Iasi, Romania. After data quality check to avoid artifacts or non-convective echoes, the authors discuss temporal (yearly, monthly, hourly) and spatial distribution of convective cells. Then, storms properties like mean displacement, VIL and other cell attributes are investigates by statistical analysis. Finally, the authors investigate relationships between basin-averaged number of storms and observed precipitation and spatial and temporal variability using EOF technique. Given duration of the observed data, ranging from 2003 to 2017, the topic addressed in the proposed work is valuable. Nevertheless, the authors report a basic statistical analysis, providing mean values and other statistical

moments with very few physical explanations or interpretations. Most of the authors' findings (e.g. storm annual, monthly or hourly behaviour) confirm well-known results in previous work (Goudenhoofdt and Delobbe, 2013). The language is poor, quite unclear, with several misspellings and it needs a deep revision. For example at page 3 line 28: "The Prut River basin is situated in the coverage are of the Weather...", or at page 4 line 14 "1st of May 2003" (repeated at page 18).

Hereafter, specific comments for each Section, Table and Figures are reported.

Section 4.1 Here, annual, monthly and hourly storm distribution is analysed. A description of statistical results is reported, confirming well-known storm characteristics (convection on May-August and during afternoon or evening). But if we wish to move further some questions are to to be answered: - are there any Prut basin peculiarities respect other places? - on 2016 an abrupt increase has been recorded: is there any explanation? - As mentioned, the year 2003 was peculiar: it is worth providing deeper explanations (synoptic conditions, atmospheric circulation anomalies over Europe) - The maximum number of storm was reached on 2016: did it happen only in Prut basin? The summer 2016 was quite rainy in large part of Europe.

Section 4.2 There are years with relatively uniform distribution of storm, while other years "hot-spots" appear: is there any link to synoptic atmospheric circulation? How can we explain this behaviour?

Section 4.3 Here, storm properties are investigated. First of all, average values are reported like the average speed, average distance: how many are the significant figures (e.g. 34.7 km and 11.4 m s-1 line 29 page 7)? The spread of sample data should be taken in account. Again, there are inter-annual variation in average value: an attempt to explain these results is needed. The authors divide storm in groups: how the criteria adopted to group data are chosen is unclear. Different storm speeds are found depending on direction: the relationship between this finding and atmospheric circulation should be investigated.

Interactive
comment

[Figure]

Section 4.4 In this section, storms and related precipitations over Prut basin are investigated. This is section the most unclear in the paper. Which is the aim of this analysis? Is it to investigate cloud precipitation efficiency? Why are weather radar-based QPEs not used? The spatial resolution of CRU dataset used is so large (50 km) that it is hard to correlate with phenomena like storm (few kilometers extent). Is it the trend in precipitation – number of storm statistically meaningful? A possible physical explanation for this result should mentioned, at least.

Section 4.5 Is the increase in the period shown by EOF analysis statistically meaningful? Figure 14 shows that the second EOF on 2013 differs from other years: is there an explanation?

Table 1

According to variables sampling distribution, median estimator is recommended as more robust than the average. The significant figures must be checked according to data spread.

Table 2

Please, check Group 3 column.

Figure 6-7 To compare year-by-year the same colour scale, number of classes and classes values for each year is recommended.

---

## Referee Comment (RC2) · Anonymous Referee #2 · 11 Feb 2019

General aspects: This manuscript describes some aspects of a 15 year convective storm climatology for the Prut River Basin in South-Eastern Romania. The climatology is based upon continuous radar measurements. The availability of homogeneous radar data and the long time series is impressive, and well described in the manuscript. The observations confirm other observations of deep convection all over Europe. However, there are a number of open questions and almost no physical interpretation of the observations are given. Over all, the manuscript requires a major revision.

Specific remarks: Page 3, line 11: what is so special about the Prut River Basin? Is convection over other parts or Romania similar? Or, was just the long time series of observations the reason for selecting this area?

Page 5, line 8: what is about merging or splitting of cells? Has it taken in account?

[Figure]

Section 4.3: The frequency of storm duration, speed, distance is certainly not normal distributed, in this case other parameters than average values are more appropriate. It would be worth to show distributions, maybe even showing the distributions for the three reflectivity groups. The 2D histograms require some discussion about the mesoscale background of the area. Is there any typical mesoscale or synoptic scale flow pattern in the region for the summer season? Neither Fig. 9 nor Fig 10 show a clear signal.

Section 4.4: Maybe my statistical knowledge is not sufficient, but I don't see any upward or downward trend in Fig. 11 and 12.

Section 4.5: Is there any physical explanation for the analysis of the EOF's? Is it possible to relate the inter annual variations to different weather patterns?

Section 5, page 11, line 3: What makes some parts of the Basin different from other parts? What causes the hot spots? It would be worth to discuss also the mountainous region west of the Prut Basin. Line 15: Is there any physical explanation for preferred travel directions?

---

## Author Comment (AC1) · 30 Mar 2019

We greatly appreciate the Referee 1 comments. These were very valuable in improving the manuscript.

Below are the authors responses.

Section 4.4 of the manuscript, named "Large scale drivers of convective storms in the Prut River basin", was added to present the physical interpretation approach suggested by the reviewer. Sea level pression and CAPE were extracted from the ERA5 reanalysis, and used to identify the large-scale drivers of local convective storms from the Prut River basin. Therefore, it was highlighted that low values of sea level pressure over the Balkan peninsula and Black Sea region are important dynamical drivers of

convective storms in the analysed area. Furthermore, through the correlation between the monthly number of convective systems in the Prut basin and CAPE, one can better identify the statistical signature of atmospheric fronts associated to the high convective activity in the Prut basin.

The language has been revised to make the manuscript much clearer to the reader.

Regarding the referee's comments on section 4.1, we underline that the statistical results reported herein are important for the readers. It is not the scope of this article to investigate any differences of convective storm characteristics between Prut river basin and other places. Also, we believe that the general physical interpretation given in Section 4.4 provides the necessary information regarding the peculiarities of the convective storm in the basin to the reader.

The spatial distribution (uniform or hot-spots) of the convective storms could be related to mesoscale flows and/or the distribution of convection parameters like CAPE. The scope of this paper is to present the distribution of convective storms at basin to regional scale in order to provide a simple but practical information to the readers. To detail the different yearly distribution, an additional deeper mesoscale analysis is needed, which can be solely the subject of a whole paper.

Storm properties section was corrected taking into account the spread of the data. To measure the central tendency, the mean was replaced with the median estimator, being more robust than the average. The inter-annual variation is given by the dynamic large-scale drivers of convective storms in the Prut River basin. The convective storms were divided into groups considering the overall median values and the radar parameters characteristic to severe weather. Storm speed versus storm direction is not necessarily given by the large-scale circulation, but can be influenced by local mesoscale condition. This investigation requires deeper analysis, which is not the aim of this paper.

Section 4.4 was rewritten to highlight the large-scale drivers of convective storms in the Prut river basin.

[Figure]

Section 4.5 was removed from the manuscript.

Table 1 was updated according to the reviewer's comments; the median estimator was used.

Table 2 was checked and corrected for misspellings.

Figures 6 and 7 were replotted, and the same colour scale, number of classes and classes values for each year were used. Also, the Prut river basin was highlighted in these figures.

---

## Author Response (AR1)

We greatly appreciate the Referee 1 comments. These were very valuable in improving the manuscript.

Below are the authors responses (blue text).

5  The manuscript investigates convective activity 15-years climatology in Prut basin, Ro- mania analysing a dataset derived from Storm Cell Identification and Tracking (Johnson et al., 1998) applied to the WSR-98D weather radar located in Barnova, Iasi, Romania. After data quality check to avoid artifacts or non-convective echoes, the authors discuss temporal (yearly, monthly, hourly) and spatial distribution of convective cells. Then, storms properties like mean displacement, VIL and other cell attributes are investigates by statistical analysis. Finally, the authors investigate relationships between basin-

10  averaged number of storms and observed precipitation and spatial and temporal variability using EOF technique. Given duration of the observed data, ranging from 2003 to 2017, the topic addressed in the proposed work is valuable.

Nevertheless, the authors report a basic statistical analysis, providing mean values and other statistical moments with very few physical explanations or interpretations. Most of the authors' findings (e.g. storm annual, monthly or hourly behaviour) confirm well-known results in previous work (Goudenhoofdt and Delobbe, 2013).

Section 4.4 of the manuscript, named "Large scale drivers of convective storms in the Prut River basin", was added to present the physical interpretation approach suggested by the reviewer.
Sea level pression and CAPE were extracted from the ERA5 reanalysis, and used to identify the large-scale drivers of local convective storms from the Prut River basin.

20  Therefore, it was highlighted that low values of sea level pressure over the Balkan peninsula and Black Sea region are important dynamical drivers of convective storms in the analysed area. Furthermore, through the correlation between the monthly number of convective systems in the Prut basin and CAPE, one can better identify the statistical signature of atmospheric fronts associated to the high convective activity in the Prut basin.

The language is poor, quite unclear, with several misspellings and it needs a deep revision. For example at page 3 line 28: "The Prut River basin is situated in the coverage are of the Weather...", or at page 4 line 14 "1st of May 2003" (repeated at page 18).

30  The language has been revised to make the manuscript much clearer to the reader.

Hereafter, specific comments for each Section, Table and Figures are reported.

Section 4.1 Here, annual, monthly and hourly storm distribution is analysed. A description of statistical results is reported, confirming well-known storm characteristics (convection on May-August and during afternoon or evening). But if

35  we wish to move further some questions are to to be answered: - are there any Prut basin peculiarities respect other places? - on 2016 an abrupt increase has been recorded: is there any explanation? - As mentioned, the year 2003 was peculiar: it is worth providing deeper explanations (synoptic conditions, atmospheric circulation anomalies over Eu- rope) - The maximum number of storm was reached on 2016: did it happen only in Prut basin? The summer 2016 was quite rainy in large part of Europe.

40

Regarding the reviewer's comments on section 4.1, we underline that the statistical results reported herein are important for the readers. It is not the scope of this article to investigate any differences of convective storm characteristics between Prut

river basin and other places. Also, we believe that the general physical interpretation given in Section 4.4 provides the reader with the necessary information regarding the peculiarities of the convective storms in the basin.

Section 4.2 There are years with relatively uniform distribution of storm, while other years "hot-spots" appear: is there any link to synoptic atmospheric circulation? How can we explain this behaviour?

The spatial distribution (uniform or hot-spots) of the convective storms could be related to mesoscale flows and/or the distribution of convection parameters like CAPE. The scope of this paper is to present the distribution of convective storms at basin to regional scale in order to provide a simple but practical information to the readers. To detail the different yearly distribution, an additional deeper mesoscale analysis is needed, which can be solely the subject of a whole paper.

Section 4.3 Here, storm properties are investigated. First of all, average values are re- ported like the average speed, average distance: how many are the significant figures (e.g. 34.7 km and 11.4 m s-1 line 29 page 7)? The spread of sample data should be taken in account. Again, there are inter-annual variation in average value: an attempt to explain these results is needed. The authors divide storm in groups: how the criteria adopted to group data are chosen is unclear. Different storm speeds are found de- pending on direction: the relationship between this finding and atmospheric circulation should be investigated.

Storm properties section was corrected taking into account the spread of the data. To measure the central tendency, the mean was replaced with the median estimator, being more robust than the average. The inter-annual variation is given by the dynamic large-scale drivers of convective storms in the Prut River basin. The convective storms were divided into groups considering the overall median values and the radar parameters characteristic to severe weather. Storm speed versus storm direction is not necessarily given by the large-scale circulation, but can be influenced by local mesoscale condition. This investigation requires deeper analysis, which is not the aim of this paper.

Section 4.4 In this section, storms and related precipitations over Prut basin are investigated. This is section the most unclear in the paper. Which is the aim of this analysis? Is it to investigate cloud precipitation efficiency? Why are weather radar-based QPEs not used? The spatial resolution of CRU dataset used is so large (50 km) that it is hard to correlate with phenomena like storm (few kilometers extent). Is it the trend in precip- itation – number of storm statistically meaningful? A possible physical explanation for this result should mentioned, at least.

Section 4.4 was rewritten to highlight the large-scale drivers of convective storms in the Prut river basin.

Section 4.5 Is the increase in the period shown by EOF analysis statistically meaningful? Figure 14 shows that the second EOF on 2013 differs from other years: is there an explanation?

Section 4.5 was removed from the manuscript.

Table 1

According to variables sampling distribution, median estimator is recommended as more robust than the average. The significant figures must be checked according to data spread.

Table 1 was updated according to the reviewer's comments; the median estimator was used.

Table 2

Please, check Group 3 column.

Table 2 was checked and corrected for misspellings.

10   Figure 6-7 To compare year-by-year the same colour scale, number of classes and classes values for each year is recommended.

Figures 6 and 7 were replotted, and the same colour scale, number of classes and classes values for each year were used.

Also, the Prut river basin was highlighted in these figures.

We greatly appreciate the Referee 2 comments. These were very valuable in improving the manuscript.

Below are the authors responses (blue text).

General aspects: This manuscript describes some aspects of a 15 year convective storm climatology for the Prut River Basin in South-Eastern Romania. The climatology is based upon continuous radar measurements. The availability of homogeneous radar data and the long time series is impressive, and well described in the manuscript. The observations
10  confirm other observations of deep convection all over Europe. However, there are a number of open questions and almost no physical interpretation of the observations are given. Over all, the manuscript requires a major revision.

Section 4.4 of the manuscript, named "Large scale drivers of convective storms in the Prut River basin", was added to present the physical interpretation approach suggested by the reviewer.
15  Sea level pression and CAPE were extracted from the ERA5 reanalysis, and used to identify the large-scale drivers of local convective storms from the Prut River basin.

Therefore, it was highlighted that low values of sea level pressure over the Balkan peninsula and Black Sea region are important dynamical drivers of convective storms in the analysed area. Furthermore, through the correlation between the monthly number of convective systems in the Prut basin and CAPE, one can better identify the statistical signature of
20  atmospheric fronts associated to the high convective activity in the Prut basin.

Specific remarks: Page 3, line 11: what is so special about the Prut River Basin? Is convection over other parts or Romania similar? Or, was just the long time series of observations the reason for selecting this area?

It is not the scope of this article to investigate any differences of convective storm characteristics between Prut river basin and
25  other parts of Romania. The authors want to highlight a method of deriving storm climatology at basin to regional scale, given a long time series of radar observations.

Page 5, line 8: what is about merging or splitting of cells? Has it taken in account?

Merging or splitting cells were not considered herein, as a general convective storm climatology was envisaged. The
30  suggested approach is scientifically challenging, but is not the scope of this paper.

Section 4.3: The frequency of storm duration, speed, distance is certainly not nor- mal distributed, in this case other parameters than average values are more appropriate. It would be worth to show distributions, maybe even showing the
35  distributions for the three reflectivity groups. The 2D histograms require some discussion about the mesoscale background of the area. Is there any typical mesoscale or synoptic scale flow pattern in the region for the summer season? Neither Fig. 9 nor Fig 10 show a clear signal.

To measure the central tendency, the mean was replaced with the median estimator, being more robust than the average. The mesoscale background related to Fig. 9 and 10 is worth being investigated, but this would lead to an extended specialized
40  analysis, and is not the aim of this paper which intends to present an approach to derive a general storm climatology based mainly on radar data.

Section 4.4: Maybe my statistical knowledge is not sufficient, but I don't see any upward or downward trend in Fig. 11 and 12

Section 4.4 was rewritten to highlight the large-scale drivers of convective storms in the Prut river basin.

Section 4.5: Is there any physical explanation for the analysis of the EOF's? Is it possible to relate the inter annual variations to different weather patterns?

Section 4.5 was removed from the manuscript.

Section 5, page 11, line 3: What makes some parts of the Basin different from other parts? What causes the hot spots? It would be worth to discuss also the mountainous region west of the Prut Basin. Line 15: Is there any physical explanation for preferred travel directions?

The spatial distribution (uniform or hot-spots) of the convective storms could be related to mesoscale flows and/or the distribution of convection parameters like CAPE. The scope of this paper is to present the distribution of convective storms at basin to regional scale in order to provide a simple but practical information to the readers. To detail the different yearly distribution, including the analysis of mountainous convection, an additional deeper mesoscale analysis is needed, which can be solely the subject of a whole paper.

[revised manuscript text omitted]